

# Incorporating Wind Sheltering and Sediment Heat Flux into 1-D Models of Small Boreal Lakes: A Case Study with the Canadian Small Lake Model V2.0

Murray D. MacKay[1]

[1]Science and Technology Branch, Environment and Climate Change Canada, Toronto, M3H5T4, Canada

*Correspondence to*: Murray D. MacKay (murray.mackay@canada.ca)

**Abstract**  Lake models are increasingly being incorporated into global and regional climate and numerical weather prediction systems.  Lakes interact with their surroundings through flux exchange at their bottom sediments and with the atmosphere at the surface, and these linkages must be well represented in fully coupled prognostic systems in order to completely elucidate the role of lakes in the climate system.  In this study schemes for the inclusion of wind

sheltering and sediment heat flux simple enough to be included in any one dimensional lake model are presented. Example simulations with the Canadian Small Lake Model show improvements in surface wind driven mixing and temperature in summer, and a reduction of the bias in the change in heat content under ice compared with a published simulation based on an earlier version of the model.

**1. Introduction**

The surface roughness of a small lake or reservoir nearly always contrasts sharply with that of its terrestrial surroundings, and will thus be associated with a different atmospheric boundary layer than would be found on shore. The new boundary layer does not develop instantaneously over the entire lake: as the atmospheric flow encounters a sudden change (generally a decrease) in roughness at the shoreline, an internal boundary layer (IBL) develops with a

transition region whose properties, including key lake mixing parameters like wind stress, vary with fetch.  In fact for sufficiently small lakes this transition region might occupy the entire surface area: wind stress varies with downstream distance and an equilibrium boundary layer never forms.  In addition, for elongated lakes and reservoirs, the net response to wind forcing will vary, depending on whether the wind is along or across the primary axis.  The impact of fetch – varying surface wind speed due to the presence of IBLs on lake evaporation (*e.g.*

Venalainen et al. 1998) and gas flux (*e.g.* Kwan and Taylor 1994) has been considered, but the effect of a sudden change in aerodynamic roughness on lake mixing does not appear to have been thoroughly examined in the literature.

On the other hand, the relationship between lake area and mixing has been investigated. Mazumder and Taylor

(1994) related both lake size and water clarity to epilimnion depth, and found linear relationships between epilimnion depth and log fetch (where fetch is taken as the square root of lake surface area) for different



transparency classes. Fee et al. (1996) also investigated the effects of lake size and water clarity on mixed layer depths, and found that lake size was the more important factor for determining mixed layer depth in a set of Canadian shield lakes, though transparency modulated this response for smaller lakes. The influence of fetch on epilimnion depth cannot be unambiguously attributed to variations in surface stress, since lake size influences

seiching and thus shear production of turbulence at the thermocline which can also contribute to mixed layer deepening (*e.g.* Gorham and Boyce 1989), but these studies do hint at the possible importance of IBLs and fetch – varying surface stress on lake hydrodynamics. Most 1-dimensional lake models assume that turbulent flux exchange with the atmosphere (heat and moisture as well as momentum) takes place under equilibrium conditions such as is assumed, for example, when employing Monin – Obukhov similarity theory.

Another physical process, particularly relevant for ice covered lakes, is the sediment heat flux. This flux arises because lake sediment, especially in shallow littoral zones, warms under the influence of penetrating radiation (as well as thermal contact with warm water) during summer and releases this heat during the ice cover season. Rizk *et al.* (2014) report results from numerous studies that found sediment heat fluxes on the order of a few $Wm^{-2}$,

generally larger during early winter and tapering off as winter progresses. This is insignificant compared with other energy fluxes during the open water season and probably within the observational uncertainty of meteorological forcing (not to mention the uncertainty in process parameterization) in numerical modelling studies. During the ice cover season, however, energy fluxes into the lake are small and this source can become important for many applications. For example, sediment heat fluxes have been linked to basin scale circulations in ice covered lakes

(Kirillin *et al.* 2012, Rizk *et al.* 2014) which can strongly influence the thermal structure and distribution of dissolved oxygen and nutrients in deep waters. The impact of sediment heat flux on surface conditions such as ice thickness and ice phenology is generally assumed small, except perhaps for very shallow lakes, though this has not been thoroughly examined in the literature

The Canadian Small Lake Model (CSLM; MacKay 2012, MacKay *et al.* 2017) is a 1-dimensional thermodynamic lake scheme developed for coupling within global and regional climate and numerical weather prediction systems. This model computes the complete nonlinear surface energy balance in a thin layer (5 cm), then solves the heat equation on a uniform finite difference grid throughout the column. Short wave radiation extinction follows Beer's Law for both visible and near – infrared bands. A diurnal surface mixed layer is simulated based on an integrated

turbulent kinetic energy (TKE) approach, with a variety of well-known sources and sinks of TKE parameterized. The seasonal thermocline arises naturally as a result of the daily excursions of the surface mixed layer. Congelation (*i.e.* black) ice forms when the energy balance in a layer is sufficiently negative to cool it below 0 ºC. Snow (*i.e.* white) ice forms when the weight of the overlying snowpack is sufficient to crack the ice and allow lake water to flood a layer of snow. The snow itself is represented with the complete snowpack parameterization component of

the Canadian Land Surface Scheme (Verseghy and MacKay, 2017). Both fractional ice cover and fractional snow – on – ice are permitted. The current formulation of the model does not account for wind sheltering or sediment heat fluxes. To begin to address these shortcomings this study proposes two new schemes which are simple and flexible





enough to be easily incorporated into any 1-D lake model. Below these schemes are fully developed from theoretical considerations and some example CSLM simulations demonstrating their impacts are presented.

## 2. Fetch – Varying Wind Stress and Mixing in Small Lakes

In the analysis that follows we consider for simplicity the case of neutral atmospheric stability with no surface heat or vapour flux. Under steady state conditions over horizontally homogeneous surfaces we expect the wind profile to be given by the familiar logarithmic form:

$$\bar{u}(z) = \frac{u_*}{k} \ln\left(\frac{z}{z_0}\right),$$

where $\bar{u}$ is the mean wind profile, $k$ is the von Karman constant, $z_0$ is the surface roughness, and $u_*$ is the surface

friction velocity of the air, which is related to surface stress by $\tau = \rho u_*^2$ where $\rho$ is the density of air. Thus upstream of a shoreline over a (likely vegetated or urbanized) terrestrial landscape we find (adopting the notation of Jensen, 1978)

$$\bar{u}(z) = \frac{u_-}{k} \ln\left(\frac{z}{z_-}\right),$$

and far downstream of the shoreline under equilibrium conditions over a very large lake we expect

$$\bar{u}(z) = \frac{u_+}{k} \ln\left(\frac{z}{z_+}\right),$$

where -,+ subscripts refer to upstream and far downstream quantities respectively. In what follows the ratio of the upstream to downstream roughness lengths, characterized by

$$M = \ln\left(\frac{z_-}{z_+}\right)$$

is an important parameter governing the response of the surface stress due to the sudden change in surface roughness at the shoreline. Over lakes the roughness elements are provided by surface waves, and thus roughness is not static but rather a function of wind speed; nevertheless, we shall consider it $O(10^{-3} m)$ in what follows. Common terrestrial surfaces are forest ($z_- = 1.0\, m$ nominally), shrubland ($z_- = 10^{-1} m$ nominally), and grassland ($z_- = 10^{-2} m$ nominally), which yield $M = 6.9, 4.6,$ and $2.3$ respectively.


The development of IBLs due to discontinuities in surface roughness over rigid surfaces has been studied for many years. This might not have been the case: as pointed out by Jensen (1978), in the planetary boundary layer the downstream equilibrium surface stress ($\tau_+ = \rho u_+^2$) under realistic conditions is different from the upstream surface





stress ($\tau_- = \rho u_-^2$) by at the very most a factor of three. If the transition from upstream to downstream conditions were simply monotonic, then Jensen speculates that there would be little interest in the development of IBLs. For most purposes a simple interpolation rule could be developed to distribute wind stress over a surface, if one were to bother at all. However experience shows that this transition is not so simple. For atmospheric flow from a rough to

a smooth surface, both experimental and theoretical results indicate that surface stress drops suddenly across the roughness transition before asymptotically approaching its new equilibrium value (*e.g* Garratt 1990). On the other hand, for flow from a smooth surface to a rough surface the surface stress initially "overshoots" the final equilibrium value before slowly asymptoting to the final value, though this situation seems less relevant for our purposes as most lakes are situated within environments that are usually rougher than the water surface.

Panofsky and Townsend (1964) developed a simple, approximate analytical model describing this phenomenon and Bradley (1968) showed that this model qualitatively captures the observed behaviour over the surfaces he examined, though he did find that the observed transition region was smaller than predicted. Panofsky and Townsend solve for a surface stress parameter given by (again adopting the notation of Jensen, 1978):

$S = (u_- - u_0)/u_-$     (1),

where $u_0 = u_0(x)$ is the surface friction velocity (of air) and is a function of fetch, *x*, with $x = 0$ representing the location of the roughness discontinuity (in our case the shoreline). Panofsky and Townsend argue that *S* can be approximated by:

$$S \approx \frac{M}{\ln\left(\dfrac{d}{z_+}\right) - 1}$$     (2),

where *d* is the depth of the IBL. They develop a relation between boundary layer depth *d* and fetch which can be solved iteratively, and substituted into Eq. (2) to solve for *S* as a function of fetch. From Eq. (1) we can then evaluate the ratio of the over lake wind stress to the upstream terrestrial surface wind stress as

$$\frac{\tau_0}{\tau_-} = (1 - S)^2$$     (3).

Jensen (1978) suggests a different approach to this problem. He argues the depth of the IBL can be solved from:

$\dfrac{d}{z_+}\left(\ln\left(\dfrac{d}{z_+}\right) - 1\right) = A\dfrac{x}{z_+} - 1$,

for some constant *A*, and that the wind stress ratio becomes

$$\frac{\tau_0}{\tau_-} = (1 - S')^2$$     (4),





where

$$S' = \frac{M}{\ln\left(\dfrac{d}{z_+}\right)}.$$

Before examining the results of these models for forest, shrubland, and grassland environments, we first revisit the
experiments of Bradley (1968) in order to gain some insight into the models. Bradley examined the flow transition
from a rough surface (created with a mesh of wire spikes: $z_- = 2.5 \times 10^{-3}$ m) to a smooth surface (runway tarmac
embedded with sand and small pebbles: $z_+ = 2.0 \times 10^{-5}$ m), and *vice versa*, in an experiment at an airfield in New
South Wales, Australia, yielding $M = 4.8$ – approximately the value for our shrubland environment. IBL depths and
stress ratios $\tau_0/\tau_-$ as a function of fetch are indicated in Fig. 1 for the Panofsky – Townsend model (solid) and Jensen
($A=1$ thin dash; $A=2$ thick dash) models. Jensen (1978) reports that for the smooth to rough transition case, setting $A$
$= 1$ reproduces the observed Bradley data very well, and yields results that are nearly identical to the theoretical
approach of Rao et al. (1974) which is based on a second order turbulence closure scheme. However, Fig. 1
suggests that for the rough to smooth case, setting $A = 2$ may be equally appropriate based on the limited data from
Bradley (maximum fetch was only about 12 m). In either case the results from Jensen appear to better represent the
observed data than does the Panofsky – Townsend approach. In addition, the free parameter $A$ could be adjusted
should more observed data become available, giving this approach some appeal.

The final downstream equilibrium surface stress $\tau_+$ (or equivalently $u_+$) is not discussed by Panofsky and Townsend,
but Jensen finds that

$$\frac{\tau_+}{\tau_-} = \left[1 - \frac{M}{\ln(Ro)}\right]^2 \qquad (5),$$

where $Ro$ is the downstream surface Rossby number given by

$$Ro = \frac{G}{f z_+},$$

$G$ is the geostrophic wind speed, and $f$ is the Coriolis parameter. Thus for a given value of $Ro$, Eq. (5) can be used to
recast results of the Panofsky – Townsend and Jensen models in terms of the approach to equilibrium of surface
stress ($\tau_0/\tau_+$) by simply dividing Eq. (3) or Eq. (4) by Eq. (5). For midlatitude lakes, we take $f = 10^{-4}$ s$^{-1}$ and $G = 10$
ms$^{-1}$. Results for lakes within forest (M=6.9; black curves), shrubland (M=4.6; blue curves) and grassland (M=2.3;
red curves) environments are shown in Fig. 2. Fig. 2a highlights the fact that IBL depth is not a function of $M$ in the
Jensen model (dashed curves) though it is obviously a strong function of $A$. Observed IBL depth data would help





constrain the value of *A*, and would be much easier to measure than surface stress – especially over open water for a sufficient range in fetch. Fig. 2b shows that an IBL in equilibrium with the downstream (*i.e.* over water) roughness has not been achieved for any terrestrial surface, even after 2 km, though not surprisingly the approach to equilibrium is much faster for the grassland case than for the forested case.

The problem at hand is to estimate a mean surface wind stress value for small lakes, given that they are embedded within terrestrial environments of vastly different surface roughness, and will thus be subject to an internal boundary layer that is almost certainly not in equilibrium with the lake surface. Fig. 2 shows that the actual surface stress $\tau_0$ is a function of fetch and always less than the equilibrium value $\tau_+$. A reasonable approach to estimate a lake – averaged surface wind stress is the following. Select a model (*e.g.* Jensen with A=2) and landcover type (*e.g.*

shrubland) and follow the appropriate curve in Fig. 2b (in this case the thick blue dash). This represents the (normalized) wind stress as a function of fetch. Simply averaging this function across the length of the lake in the direction of the wind gives the appropriate normalized mean wind stress. It is clear that for irregularly shaped lakes different wind directions will lead to different results: the mean wind stress for elongated lakes with the wind blowing along the major axis of the lake will be larger than for winds blowing across the major axis. Thus a simpler

approach would be to compute this ratio for the maximum lake fetch, yielding an upper limit on the surface stress.

It is worth pointing out that Vickers and Mahrt (1997) found that surface drag *decreased* with fetch, because younger, growing waves are associated with larger surface drag than older waves (unless they are breaking). In reality both processes may be occurring, but it is clear that for the smallest lakes the mechanism described here will dominate. Stepanenko *et al.* (2014) have proposed that excessive drag in the 1-D model LAKE for a simulation of

Lake Valkea – Kotinen, a 4.1 ha boreal lake in southern Finland, can be compensated for by partitioning atmospheric momentum flux between wave development and surface currents. Our results provide an alternative (or perhaps additional) explanation. Fetch in this lake varies from about 100 m to 400 m and the lake is surrounded by forest. Examination of Fig. 2 suggests that surface stress (or equivalently the surface drag coefficient) is only a fraction of the equilibrium value: perhaps between 25% and 50% depending on the model chosen.

We have applied this technique to a simulation of the CSLM for a small boreal lake (L239 at the Experimental Lakes Area, Canada) for 2013 – 2014 that has been described in MacKay *et al.* (2017). In that study the primary focus was on the winter season, though the simulation actually began in July. The first few days of this simulation (17 – 25 July, 2013) are illustrated in Fig. 3 where observations are indicated with black curves, the original simulation with blue curves and a modified simulation with red curves. Both 19 July and 22 July were relatively

windy days (Fig. 3a) with winds generally from the west or north-west (not shown) yielding a maximum fetch around 500 m. Thus based on Fig. 2b and given that L239 is in a forested catchment we have reduced the surface drag coefficient by 50% in the modified simulation. The impact has been an improved simulation of surface (*i.e.* 0.5 m depth) temperature (Fig. 3b) caused by a reduction is simulated diurnal mixed layer depths (Fig. 3c), especially during or shortly following the wind events.




With this approach we must keep in mind the following two caveats. Very large changes in roughness or very dense canopies are strictly speaking not consistently handled as no allowance has been made for changes in the zero plane displacement (*e.g.* Markfort *et al.* 2014). Also, unlike for rigid surfaces, steady wind over open water will generally lead to an increase in surface roughness with time. In addition, following Panofsky and Townsend we consider only

neutrally stratified boundary layers. All of these issues could likely be incorporated into the theory at the expense of increasing complexity; nevertheless, this approach should at least qualitatively describe the importance of IBLs and fetch – varying wind stress over small lakes, and appears a suitable first step.

### 3. Sediment Heat Flux

Currently the CSLM employs an adiabatic boundary condition at lake bottom. In the recent simulation of L239 mentioned above, it was found that during a 100 day ice covered period (2013-2014) the simulated change in lake heat content corresponded to a mean bias of about -2.5 W m$^2$ compared to observations (MacKay *et al.* 2017). The authors noted that this was of the same order of magnitude as the wintertime sediment heat flux found in a number of previous studies and suggested this warranted further investigation. Here we test whether the inclusion of a

simple sediment heat flux scheme can ameliorate this bias.

A simple scheme for the storage and subsequent flux of heat from lake sediments can be constructed by considering a sediment slab of fixed thickness, and uniform thermal conductivity and volumetric heat capacity. Such a scheme is simpler than some existing multilayer sediment models (Stepanenko *et al.* 2016), but has the appeal of not requiring many additional levels to be carried in global weather and climate simulations. Mean temperature in the

slab evolves due to thermal and radiative fluxes at the lake water – sediment interface in such a way as to conserve energy. The boundary condition at the base of the slab is isothermal, rather than adiabatic as is sometimes assumed (*e.g.* Stepanenko *et al.* 2016), which places a constraint on the minimum slab thickness. Note that for terrestrial (*i.e.* soil or rock) surfaces the diurnal temperature wave is believed to penetrate about 1 m and the annual temperature wave penetrates about 20 m (e.g. Carslaw and Jaeger, 1959), below which geothermal heating acts to maintain a

constant temperature gradient. A number of studies have found that temperature oscillations in lake sediments are substantially damped after only a few m, below which temperatures are essentially isothermal (*e.g.* Likens and Johnson 1969, Tsay *et al.* 1992). Here we consider a slab thickness of 10 m, with thermal properties consistent with sand. A layer of pure sand has a thermal conductivity of about 2.5 W K$^{-1}$ m$^{-1}$ (about 4 times larger than that for liquid water) and a volumetric heat capacity of about 2.13 x 10$^6$ J m$^{-3}$ K$^{-1}$ (about half that of liquid water) and these

values are adopted here. Values for clay would be similar though sediments with significant organic matter would differ. The system is solved by asserting continuity of both temperature and heat flux at the water – sediment interface (*i.e.* option 1 in Stepanenko *et al.* 2016), and a lower boundary condition temperature of 6.0 ºC.

The difficulty with applying such an approach here is that the CSLM does not include lake bathymetry information, and lake bottom is everywhere assumed to be at the mean lake depth. In fact many lakes have extensive shallow

littoral zones in which we would expect much larger shortwave (SW) insolation, and thus sediment heat flux, than in





more "bathtub" shaped lakes which nevertheless have the same mean depth. Here we propose a scheme to compute the net sediment SW insolation based on minimal bathymetric data – maximum and mean depth only.

In what follows, all lower case variables are dimensionless – scaled by appropriate length and depth scales for the lake in question. Fig. 4a shows a family of 1-parameter theoretical "hypsographs" given by

$$y = x^s,$$

where x is the (dimensionless) radial distance from lake centre, y is the (dimensionless) height from the lake bottom at maximum depth, and s is a shape factor. Theoretical, axially symmetric lakes are formed by rotating these curves around the line x=0. Thus for s=1 the lake is conical, for values of s less than 1 the lake takes on a "birdbath" shape, while for values of s greater than 1 it is more "bathtub" shaped. In the limit of very large $s$ the lake becomes a right

circular cylinder (the default shape assumed in CSLM). Birdbath lakes have extensive shallow littoral zones whose sediment would absorb much more SW radiation than bathtub lakes of the same surface area.

Reference to Fig. 4b shows that the volume of revolution of a thin disk at height $y$ is

$$dv = \pi x^2 dy,$$

so that the total volume is simply

$$v = \int_0^1 \pi x^2 dy \quad = \int_0^1 \pi y^{2/s} dy \quad = \pi \frac{s}{s+2}.$$

The dimensional volume is thus

$$V = \pi \frac{s}{s+2} L^2 H_{max},$$

where $H_{max}$ is the maximum depth and $L$ is the radius of a circular lake with the same surface area as the lake in question. We can determine the value for $s$ by equating this volume to the actual volume of the lake, or

equivalently:

$$\pi \frac{s}{s+2} L^2 H_{max} = \pi L^2 \bar{H},$$

where $\bar{H}$ is the mean lake depth. This yields

$$s = \frac{2\frac{\bar{H}}{H_{max}}}{1 - \frac{\bar{H}}{H_{max}}}.$$

25
For L239 we have $\bar{H} = 11.0 \, m$ and $H_{max} = 30.0 \, m$ so that $s = 1.16$. Fig. 4c compares the actual (normalized) hypsograph for L239 compared with our theoretical curve.

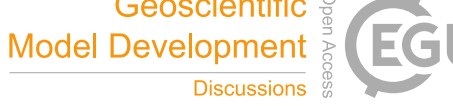



The task now is to estimate the mean absorbed SW radiation at the sediment-water interface given this lake shape profile. It is clear from Fig. 5 that if $i_0$ is the intensity of SW radiation reaching the lake surface (per unit area), then the intensity reaching the sediment at radial distance $x$ is given by

$$i_{sed}(x) = i_0 \cos\theta \, e^{-\beta(1-x^s)}$$

5   where β is the (dimensionless) short wave extinction (*i.e.* $\beta = \hat{\beta} H_{max}$ where $\hat{\beta}$ is the dimensional extinction) and 1-$x^s$ is the depth at $x$. Consider the surface area of a thin ring of radius $x$ and width *dl*. Clearly

$$da = 2\pi x \, dl,$$

so that the total radiation reaching this elemental surface is

$$i_{sed}(x) da = i_0 \cos\theta \, e^{-\beta(1-x^s)} 2\pi x \, dl$$

$$= 2\pi i_0 x e^{-\beta(1-x^s)} \, dx.$$

The total insolation of the lake sediment surface is found by integrating over all x, and the mean sediment insolation per unit lake surface area is found by dividing this integral by the lake surface area. Recalling that the dimensionless lake surface area is simply π we get:

$$\bar{\imath}_{sed} = \frac{1}{\pi} \int_0^1 2\pi i_0 x e^{-\beta(1-x^s)} dx$$

$$= i_0 \left[ 2e^{-\beta} \int_0^1 x e^{\beta x^s} dx \right]. \tag{6}$$

This has analytic solutions in terms of the gamma and incomplete gamma functions:

$$\bar{\imath}_{sed} = 2i_0 e^{-\beta} \left\{ \frac{1}{s}(-\beta)^{-\frac{2}{s}} \left[ \Gamma\left(\frac{2}{s}\right) - \Gamma\left(\frac{2}{s}, -\beta\right) \right] \right\} ; \quad s > 0, \tag{7}$$

Note that for s=0 (*ie.* a birdbath of zero depth everywhere except at the centre) the solution Eq. (7) does not apply but Eq. (6) can be integrated trivially to get

$$\bar{\imath}_{sed}(s=0) = 2i_0 e^{-\beta} \int_0^1 x e^{\beta} dx = i_0,$$

which is the correct limit. On the other hand, for very large *s* our lake becomes cylindrical and Eq. (6) can also be solved trivially. Expanding the exponential in a Taylor series we find:

$$\bar{\imath}_{sed,\infty} = \lim_{s\to\infty} \left\{ 2i_0 e^{-\beta} \int_0^1 x e^{\beta x^s} dx \right\}$$

$$= 2i_0 e^{-\beta} \lim_{s\to\infty} \left\{ \int_0^1 x \left( 1 + \beta x^s + \frac{(\beta x^s)^2}{2!} + \cdots \right) dx \right\}$$

25
$$= 2i_0 e^{-\beta} \lim_{s\to\infty} \left\{ \left[ \frac{1}{2} x^2 + \frac{\beta}{s+2} x^{s+2} + \frac{\beta^2}{2(2s+2)} x^{2s+2} + \cdots \right]_0^1 \right\}$$



$$= i_0 e^{-\beta} \,,$$

which is again the correct limit.

For L239 we have found $s=1.16$ and the (dimensionless) extinction is $\beta=27$ so that Eq. (6) or Eq. (7) becomes

$$\bar{\imath}_{sed} = 0.06 i_0$$

In other words, of the net surface SW radiation on L239, about 6% reaches the sediments leaving 94% to be absorbed by the lake water on a lake wide average. In the standard simulation assuming the entire lake is at the mean depth (11 m), the sediment insolation is only

$$\bar{\imath}_{sed} = 5.0 \times 10^{-5} i_0,$$

three orders of magnitude less. The approach taken here is to simply reduce the incoming net SW insolation at the
lake surface by 6% and apply this energy flux directly at the lake water – sediment interface. In this way we estimate the mean sediment insolation for any lake whose surface insolation is known, along with extinction, mean and maximum depths. For example, Table 1 lists sediment SW insolation fractions for a variety of shape factors $s$ for lakes with the same (dimensionless) extinction as L239.

| $s$ | $\bar{\imath}_{sed}/i_0$ |
|---|---|
| 0.1 | 0.43 |
| 0.5 | 0.13 |
| 1.0 | 0.07 |
| 2.0 | 0.04 |
| 10.0 | 0.01 |

**Table 1: Sediment shortwave insolation fractions for lakes with dimensionless shortwave extinction β=27 and various**
**shape factors $s$.**

The above approach was used in two new simulations of L239 for 2013-2014 (Fig. 6). In the standard simulation for this period (MacKay *et al.* 2017) the lake bottom was adiabatic. In order to isolate the impact of geothermal heating alone we relaxed the adiabatic boundary condition as described above but set the sediment SW insolation fraction to 0.0 (experiment X1; Fig. 6 blue curves). The second experiment (experiment X2) is identical to the first
except we set the sediment SW insolation fraction to the proposed value of 0.06 (Fig. 6 red curves). The simulated ice thicknesses (Fig. 6a) are virtually identical in the two experiments though both the sediment temperatures (dotted curves) and lowest water layer temperatures (solid curves) differ even well before ice-on (Fig. 6b). Note that the sediment heat flux is close to zero or negative (*i.e.* into the water column) throughout these simulations, with values never exceeding a few W m$^{-2}$ (Fig. 6b dashed curves, right-hand scale). During the ice cover season the geothermal
component is significant, generally about half of the total.

Finally, the impact of the new sediment heat flux scheme on lake thermal structure during the ice cover period is illustrated in Fig. 7. Ice phenology and thickness for the original and modified simulations are virtually identical (Fig. 7a). Temperature profiles in the original experiment show no sign of water column warming (Fig. 7b) – in fact a general cooling trend is evident in the top 5 m until ablation begins (after snow cover is gone – not shown). The



observations on the other hand show clear signs of warming throughout most of the column under ice (Fig. 7c). Results from the sediment heat flux experiment (X2) more closely resemble this pattern (Fig. 7d), though there are obvious deficiencies in the simulation including a more stratified region for several meters below the ice as well as a general warm bias in deeper waters and a too strongly stratified surface layer near and following ice-off.

5 The change in total heat content under ice is, however, improved. MacKay *et al.* (2017) found that between 26 November 2013 and 6 March 2014 (100 days) the simulated $1 - 10$ m water column lost $7.26$ x$10^6$ (J m$^{-2}$) corresponding to a mean heat flux of $-0.84$ (W m$^{-2}$) whereas observations suggested warming corresponding to $+1.66$ (W m$^{-2}$). Including sediment heat flux improves this. Geothermal heating alone (X1) brings the mean heat flux into the column up to $0.08$ (W m$^{-2}$), while including the radiative forcing (X2) yields $1.05$ (W m$^{-2}$) for a net bias 10 of $-0.61$ (W m$^{-2}$). These results are summarized in Table 2.

| Experiment | $\Delta Q_{1\text{-}10\,m}$ (W m$^{-2}$) | Bias (W m$^{-2}$) |
|---|---:|---:|
| Observations | 1.66 | - |
| MacKay *et al.* (2017) | -0.84 | -2.50 |
| X1 | 0.08 | -1.58 |
| X2 | 1.05 | -0.61 |

**Table 2: Observed and simulated change in 1-10 m heat content from 26 November 2013 – 6 March 2014.**

A disadvantage of the method described here is that all sediment heat flux into the lake water takes place at the mean lake depth (which is the lowest model level). Circulation processes that redistribute heat are clearly active in the observations (Fig. 7c) but are not represented in the model. In fact the only reason the simulated heat is 15 redistributed vertically is that the bottom simulated temperatures are near the temperature of maximum density (4$^\circ$ C) so that warming of these waters produced convection. But temperatures in the bottom half of the simulated lake are clearly biased warm during ice cover, and free convection is likely not taking place in reality. So while the change in total heat content under ice has improved with the new scheme, work remains for parameterizing mixing processes to appropriately redistribute heat emanating from the sediments.

20 **4. Conclusion**

Lakes are increasingly being recognized as important components of the land surface, and 1-D modeling schemes are currently being developed for inclusion in a number of climate and numerical weather prediction systems around the World. Lakes interact with their environment through flux exchange with their bottom sediments and at the surface with the atmosphere, and even a perfect lake model will perform poorly in a prognostic sense if these 25 linkages are represented poorly. However these are challenging areas of research: high quality data in lake sediments (*e.g.* temperature, thermal properties) are difficult to achieve at regional or global scales, and the process understanding itself of turbulent exchange under non-equilibrium conditions, ubiquitous for all but the largest of boreal lakes, is lacking. This study contributes to this discussion with two simple schemes that begin to represent these linkages in simple 1-D "bathtub" like models like the CSLM.


Terrestrial landscapes play a role in lake hydrodynamics through the sudden drop in aerodynamic roughness that the atmosphere encounters at the shoreline of virtually any boreal lake. While some models do make adjustments to



surface drag to account for various processes (or for "tuning"), none to our knowledge factors in the terrestrial roughness in a quantitative way. Here we propose a straightforward scheme based on earlier IBL research that begins to address this. When applied to a simulation of the CSLM we have found an improvement in the near surface temperature following two wind events a few days apart, that resulted from a more realistically simulated

diurnal mixed layer depth.

It is well known that shallow lakes or lakes with extensive shallow littoral zones may be subject to sediment heating that is significant for many applications, and several models have attempted to account for this process. While it is trivial to add a sediment layer beneath a lake in order to absorb and subsequently release heat, problems arise when

the lake model itself does not represent bathymetry. Since much of the sediment heat content arises from SW insolation at the sediment – water interface, lake models that assume uniform depth (*i.e.* constant depth set equal to the lake mean depth) would in most cases receive almost no radiative forcing at the sediment unless the lake was exceptionally shallow or clear. To account for this a scheme is proposed to estimate the actual sediment insolation by approximating any given lake as an axially symmetric lake with a shape factor determined from mean and

maximum depth data only. When applied in a simulation of the CSLM we have found an improvement in the change of lake heat content under ice cover over a 100 day period. Issues remain regarding the distribution of the sediment heating throughout the water column, as currently all heating takes place at the lake mean depth.

*Code and data availability.* Code for the CSLM and data used for its forcing and evaluation are available at

http://doi.org/10.5281/zenodo.2554524

*Author Contributions.* M.D. MacKay performed all tasks associated with this manuscript including developing the mathematical framework, coding and running the numerical model, analyzing the simulations, and writing the manuscript.


*Acknowledgements.* Mike Rennie provided temperature profile observations under ice cover for L239. As always, I am indebted to the IISD – ELA staff who have provided ongoing and invaluable support throughout this research programme.

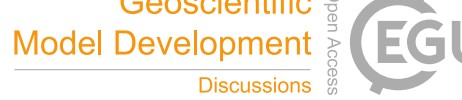

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



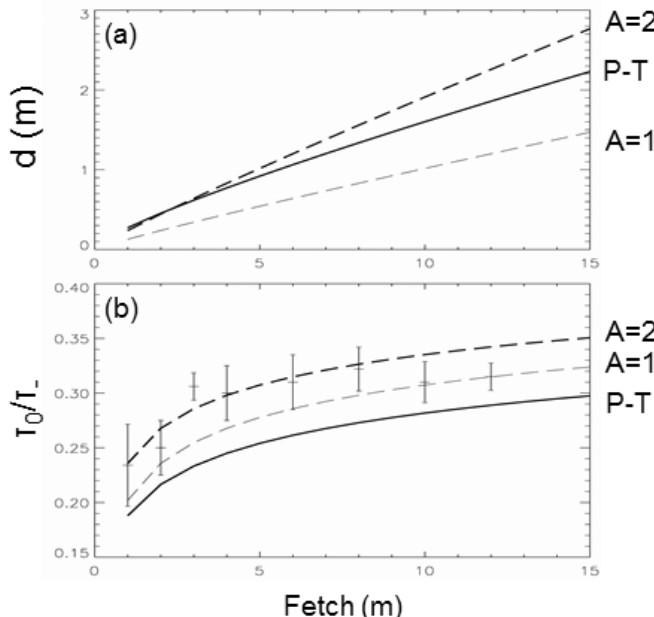

**Figure 1:** Theoretical IBL depths (a) and surface stress (b) as a function of fetch for the
rough – to – smooth transition case of Bradley (1968). Solid curves – Panofsky and
Townsend (1964) approach; dashed curves – Jensen (1978) approach with A=1 (thick dash)
and A=2 (thin dash) as indicated. Error bars in (b) represent the range of observed values
from Bradley (1968), and surface stress values are normalized by the upstream value.





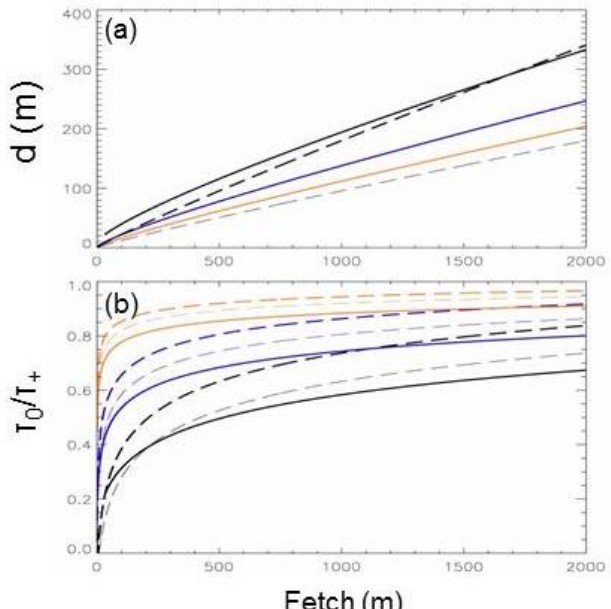

**Figure 2:** Theoretical IBL depths (a) and surface stress (b) as a function of fetch over water for a
rough – to – smooth transition (off shore flow) for forested landscapes (black curves, M=6.9);
shrubland (blue curves, M=4.6), and grassland (red curves, M=2.3) . Solid curves – Panofsky
and Townsend (1964) approach; dashed curves – Jensen (1978) approach with A=1 (thin dashed),
and A=2 (thick dashed).  Surface stress values are normalized by the theoretical equilibrium
downstream value.





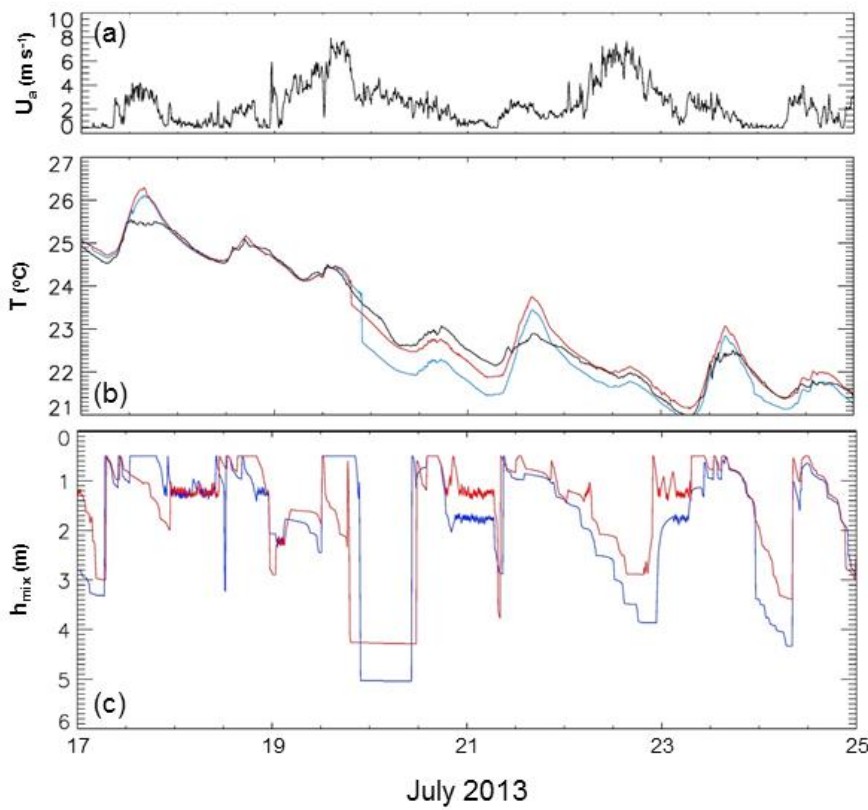

**Figure 3:** Observed 2-m wind speed (a), observed (black) and simulated 0.5 m water temperatures (b), and simulated diurnal mixed layer depths (c) for 17 – 25 July, 2013. Standard simulation results (blue curves) taken from MacKay *et al.* (2017). Modified simulation results (red curves) have surface drag coefficient reduced by 50%.





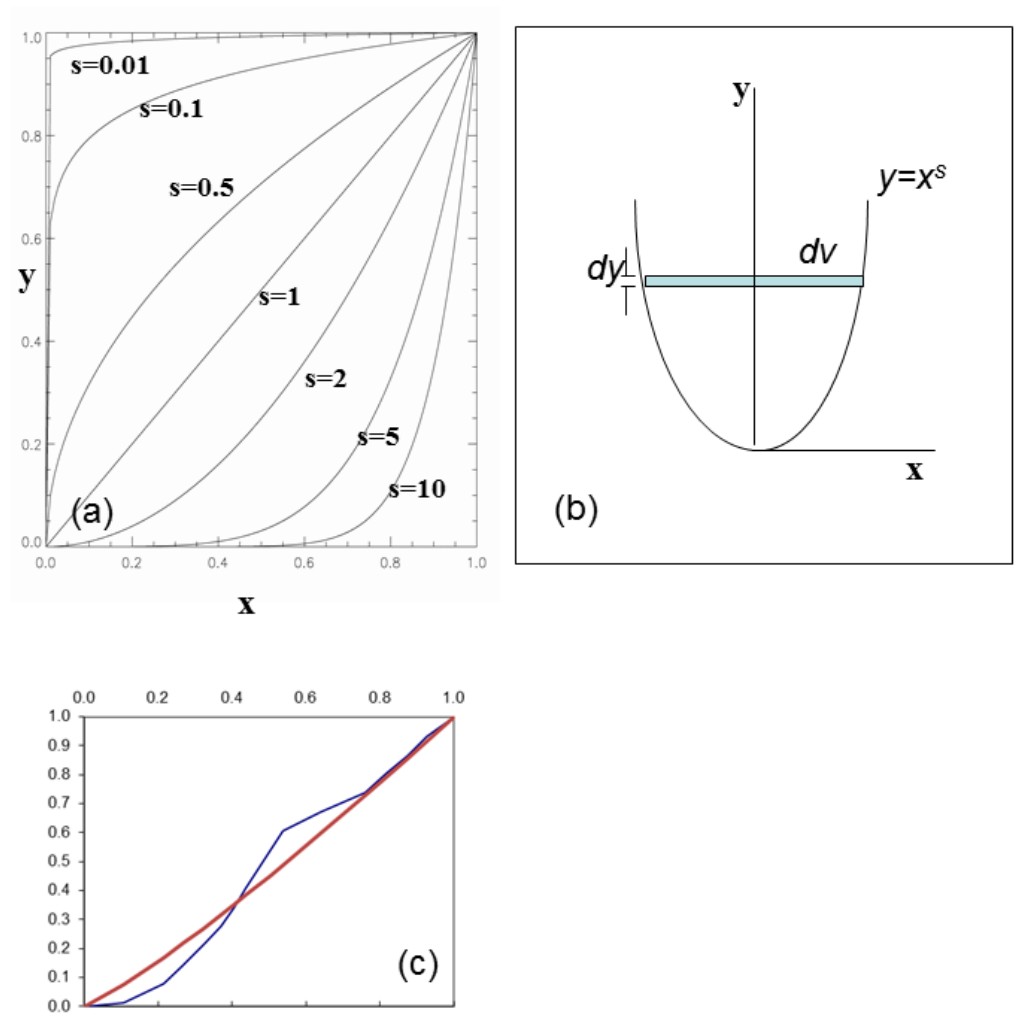

**Figure 4:** (a) Theoretical "hypsographs" for idealized axially symmetric lakes with different shape factors *s* indicated; (b) schematic to aid in determining volume of an axially symmetric lake; (c) normalized observed hypsograph (blue) and theoretical estimate (red) for L239. See text for details.



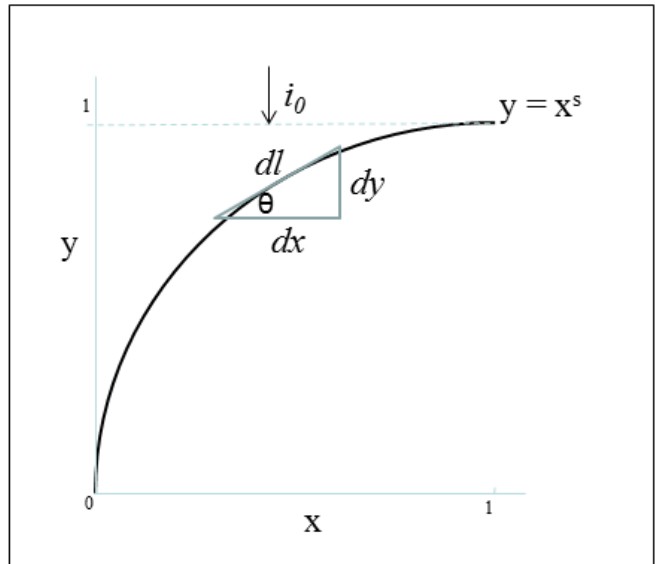

**Figure 5:** Schematic to aid in the estimation of sediment shortwave insolation for an axially symmetric
lake with a given shape factor $s < 1$.



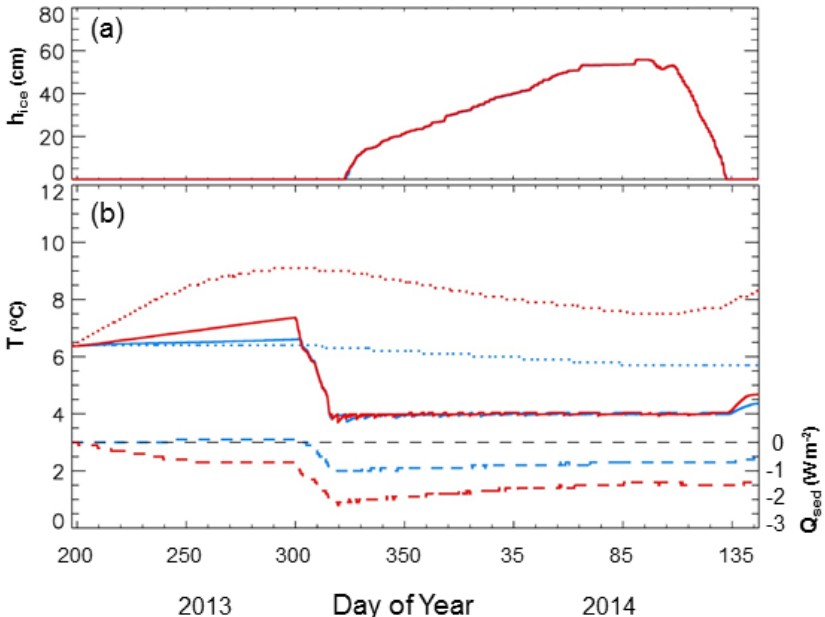

**Figure 6:** (a) Simulated ice thickness; (b) simulated sediment temperature (dotted), lowest level water temperature (solid), and sediment heat flux (dashed – right hand scale). Simulated results are from experiment X1 (blue) and X2 (red). Zero sediment heat flux (dashed black curve) is also indicated in (b).

5   The period shown covers 19 July 2013 – 25 May 2014.



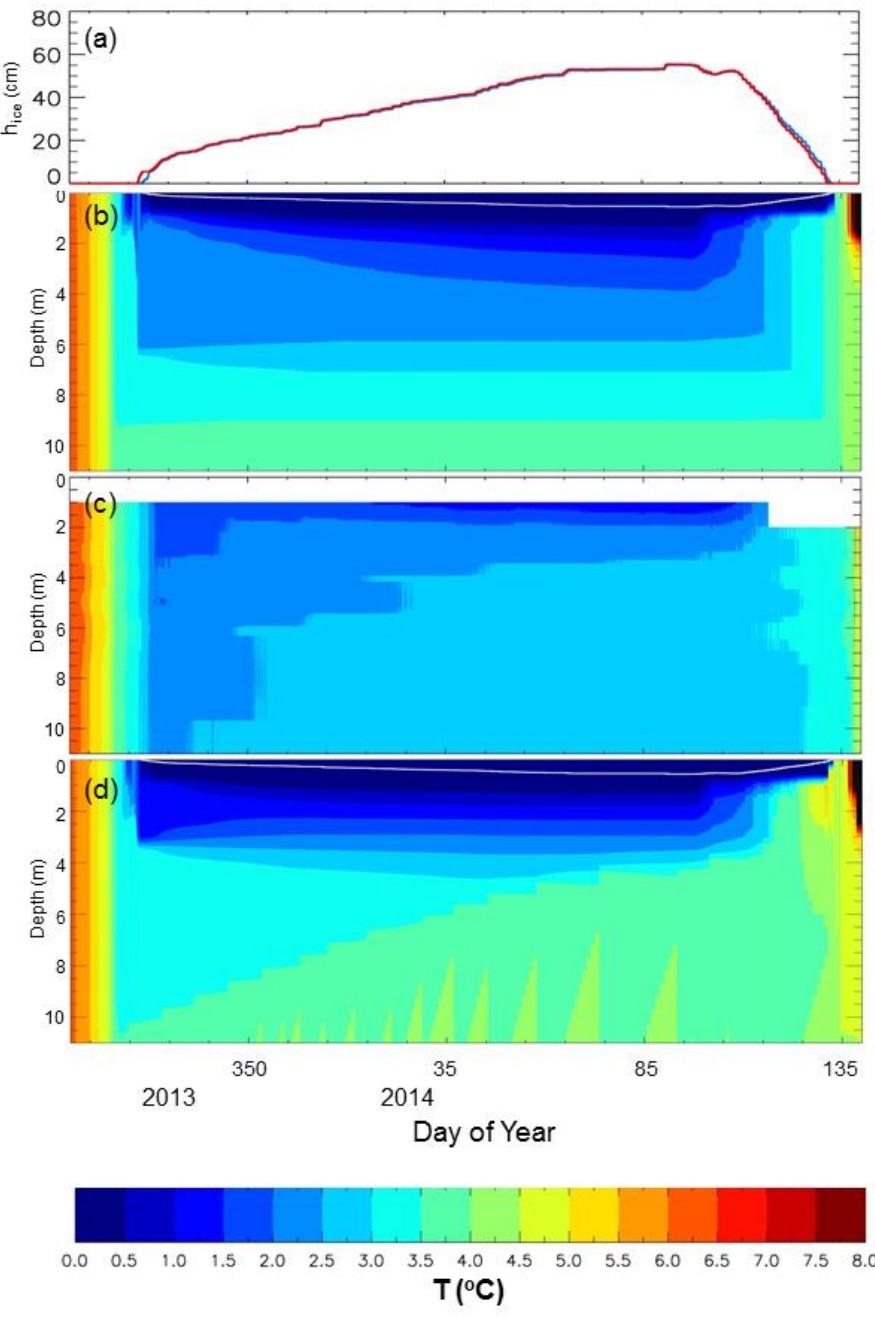

**Figure 7:** Simulated ice thickness (a), and simulated (b),(d) and observed (c) temperature profiles for 1 November 2013 – 20 May 2014. The standard simulation (MacKay *et al.* 2017) is shown in (b) and the blue curve in (a). Simulation X2 is shown in (d) and the red curve in (a).