# Peer review of "Incorporating Wind Sheltering and Sediment Heat Flux into 1-D Models of Small Boreal Lakes: A Case Study with the Canadian Small Lake Model V2.0"

_Geoscientific Model Development, 2018_

## Referee Comment (RC1) · Anonymous Referee #1 · 10 Mar 2019

The study discusses possibilities of incorporating 2-d horizontal inhomogeneities into 1-dimensional vertical models of lake thermodynamics. Two numerically inexpensive parameterizations are developed by the author and tested in the framework of the Canadian Small Lake Model (CSLM): one dedicated to accounting of the surrounding roughness on the air-lake turbulent fluxes, the other one dealing with the heat storage by the lake sediments in lakes of different morphometry.

The first parameteization reproduces the sheltering effect of rough surroundings (e.g. forest) on small lakes, based on laboratory experiments data on turbulent stress transition from a rougher to a smoother surface. According to test model runs, the parameterization improves remarkably simulation of the surface mixed layer depth in a small Canadian lake and leads to a generally better simulation of lake surface temperatures. The approach is promising for modeling small boreal lakes, sheltered by forests, which are abundant in the northern latitudes. Since the approach does not need extensive calculations, it can be potentially incorporated in lake parameterization schemes of global/regional models. Some clarifications is nesessary here, before such global-scale implementation becomes possible:

(i) While using the Coriolis-scaling for the equilibrium surface stress (P5L20 Eq. 5) seems reasonable in high latitudes, it would apparently fail in tropics. A comment on possible ways of generalization or alternative scaling is needed; (ii) the algorithm proposed at P6L5-15 and Fig. 2 works fine for a single lake and provides a nice visual demonstration of the possible corrections. However, a final formula $\tau_0/\tau_+(fetch, G, z_+ \ldots)$ would allow the reader to directly test/incorporate the parameterization in other models avoiding diagrams and discrete choices of the landcover type; (iii) P6L16: if the wave aging produces the opposite effect to the sheltering on the surface stress, an estimation of the fetch length (lake size) at which the wave age becomes more important than the sheltering effect (land-lake transition) is needed.

The demonstrated effect of the second parameterization—incorporation of the solar heating of shallow sediment—on the model output is less obvious, being probably overwhelmed by other deficiencies of 1-d modeling approach. In the presented application, the CSLM seems to underestimate the verical mixing in the upper part of the ice-covered water column. As a result, a spurious convection is produced at the lake bottom, when the solar heating of the sediment surface is added. The mathematical framework is in turn well-developed and allows easy adaptation of the algorithm to a specific modeling task. In this sense, the approach offers a potentially effective way of incorporating lake morphometry into 1d horizontally-integrated models.

Below are questions on the second part of the study:

P7L21 "The boundary condition at the base of the slab is isothermal... which places a constraint on the minimum slab thickness" It sounds not very physical and numerically problematic. Why not using a constant (geothermal or zero) flux at the sediments' base or at an infinite depth instead?

P7L28 Do the conductivity and heat capacity values refer to dry or water-saturated sand?

P7L27-30 Why the sediment thickness 10 m and the lower boundary condition temperature of 6.0 °C are chosen?

P11L8 "Geothermal heating alone (X1) brings the mean heat flux into the column up to 0.08 W m$^{-2}$ ...". Geothermal flux in this model formulation is not a result of simulations, but is artificially prescribed by setting the slab thickness and the temperature at its base. Does any evidence exist of geothermal flux of similar magnitudes in Canadian lakes?

P12L10 "Since much of the sediment heat content arises from SW insolation at the sediment – water interface ..." The statement needs some support. Contribution of turbulent heat transport from water to sediment can be at least as important as radiative heating.

minor/technical remarks:

P1L34 The references here seem slightly outdated. See Kirillin&Shatwell (2016 *Earth-Science Reviews*, 161, 179-190, https://doi.org/10.1016/j.earscirev.2016.08.008) for a discussion on the relationship between fetch/transparency and epilimnion depth.

Fig. 2: $\tau$ looks like T in the y-axis subscript.

Fig. 3: While surface stress correction resulted in generally better prediction of surface temperatures, it also produced a stronger overestimation of temperatures during daytime on 17, 21 and 23 July (calm and warm/sunny days?). Any comments on the background mechanisms?

[Figure]

---

## Referee Comment (RC2) · Anonymous Referee #2 · 27 Mar 2019

A paper "Incorporating wind sheltering ..." by Murray D. MacKay presents two improvements of a 1D Canadian Small Lake Model v2.0, namely, a correction to surface drag coefficient, representing effects of sharp roughness discontinuity at the lake-shore interface; and modification of radiation scheme simulating integral effect of shortwave radiation absorption at variable lake depth. The approaches and hypotheses involved are clearly described and are mathematically elegant. Although the observational data used to validate new parameterizations is limited, both parameterizations are worth publishing and are highly relevant to the journal scope. However, I feel the manuscript

could be hopefully improved by considering the following comments:

1) The wind-sheltering effect is parameterized using analytical models for internal boundary development over roughness boundaries, published in previous century. Since then, there have been new studies performed elucidating turbulent flow dynamics (including surface stress) over such surfaces, e.g. Large Eddy Simulation (Glazunov and Stepanenko, 2015; Kenny et al., 2017), wind tunnel experiments (Markfort et al., 2010; Markfort et al., 2014) and eddy covariance measurements (Queck et al., 2016; Barskov et al., 2017). Though, the author briefly mentions Markfort et al., 2010, I would expect more discussion on the subject.

2) The author assumes water surface momentum roughness to be of $O(10^{-3})$ m. However, from my experience this likely to be an overestimation of typical values. Could you provide more grounds on choosing this value, or check the sensitivity of the model to this parameter? The other option would be simulating $z\_0$ by Charnock formula modified with fetch-dependence, like it is done in FLake model.

3) Fig. 3, bottom panel. Do you have observation data for the mixed-layer depth, derived from temperature measurements?

Literature

Barskov K. V, Chernyshev R. V, Stepanenko V.M., Repina I.A., Artamonov A.Y., Guseva S.P., Gavrikov A. V. Experimental study of heat and momentum exchange between a forest lake and the atmosphere in winter // IOP Conf. Ser. Earth Environ. Sci. 2017. Vol. 96. No. 1. pp. 12003.

Markfort C.D., Porté-Agel F., Stefan H.G. Canopy-wake dynamics and wind sheltering effects on Earth surface fluxes // Environ. Fluid Mech. 2014. Vol. 14. No. 3. pp. 663–697.

Markfort, C. D., Perez, A. L. S., Thill, J. W., Jaster, D. A., Porté‐Agel, F., and Stefan, H. G. ( 2010), Wind sheltering of a lake by a tree canopy or bluff topography, Water

Resour. Res., 46, W03530, doi:10.1029/2009WR007759.

Kenny W.T., Bohrer G., Morin T.H., Vogel C.S., Matheny A.M., Desai A.R. A Numerical Case Study of the Implications of Secondary Circulations to the Interpretation of Eddy-Covariance Measurements Over Small Lakes // Boundary-Layer Meteorol. 2017. Vol. 165. No. 2. pp. 311–332.

Glazunov A.V., Stepanenko V.M. Large-eddy simulation of stratified turbulent flows over heterogeneous landscapes // Izv. - Atmos. Ocean Phys. 2015. Vol. 51. No. 4.

Queck R., Bernhofer C., Bienert A., Schlegel F. The TurbEFA Field Experiment—Measuring the Influence of a Forest Clearing on the Turbulent Wind Field // Boundary-Layer Meteorol. 2016. Vol. 160. No. 3. pp. 397–423.
* * *

---

## Author Response (AR1)

*Wind Sheltering:*

Response to Anonymous Referee #1

The referee points out that any estimate of the equilibrium surface stress based on a Rossby number would not be appropriate in a global model, even if it does work well for mid-latitude lakes as discussed here. In fact, on further analysis (including correcting a typo in the plotting programme) we now show that this estimate is not even appropriate for the mid-latitudes. Jensen's original notion was that equilibrium was reached when the IBL filled the entire PBL, but it is easily shown that at the latitude of our study lake this would require fetches of several thousand km. However, plotting $\tau_0/\tau_-$ (new Fig. 2) it is easily seen that values for this ratio asymptote much sooner than this – generally for fetches of 5 km or less. New text describing this has been included starting around P5 L26 (marked-up text).

The new approach is not presented in terms of a final formula as suggested by the referee (which would not be easily expressed) but a simple algorithm is now outlined in the text near P6 L15 (marked-up text).

The full impact of wave state would add an enormous complexity to the current formulation, even if individual aspects could be incorporated. We have chosen simply to point out that this is a real issue, well beyond the scope of this study, and requires further research.

Response to Anonymous Referee #2

A more fulsome discussion on recent relevant wind tunnel, large eddy simulation, and field research is now presented (starting near P7 L24, marked-up text). The important phenomena of streamline displacement and flow reattachment in the lee of the forest edge is discussed, and it is noted that the empirical surface stress model of Markfort *et al.* (2014) appears similar to what is proposed here if the distance to flow reattachment is considered.

We have found that our results are not terribly sensitive to the choice of a (constant) water surface roughness of $10^{-3}$ m. Choosing values of $10^{-2}$ m or $10^{-4}$ m changes the mean surface stress reduction factor from 0.50 to 0.59 and 0.45 respectively, leading to quantitative but not qualitative differences (see P7 L37, marked-up text).

Fig. 3 now includes observed and simulated temperature profiles for a few days between 17-25 July capturing the impact of the wind mixing events. These clearly show differences in the simulated

epilimnion temperatures and depths compared with observed. Discussion has been added (starting near P7 L15, marked-up text).

***Sediment Heat Flux:***

Response to Anonymous Referee #1

The boundary condition at the base of the sediment slab layer has been fixed isothermal at 6.0 °C for a layer 10 m thick. These values are somewhat arbitrary: there are few data beneath boreal lakes in our region to support these (nor have we been able to find any geothermal flux data below Canadian boreal lakes). Nevertheless our experience with lake L239 is that hypolimnion water temperatures rarely deviate from 4.0 °C throughout the year, so we assume the sediment base temperature should be close to this. Likens and Johnson (1969) found that data from Wisconsin (several hundred km to the south-east of our research area) show no seasonal variation in sediment temperatures below about 10 m in the lakes they examined, which motivates choosing our slab thickness. They found nearly isothermal conditions at this depth of around 6 – 8 °C. In this study, thermal properties of the sediment slab have been assumed the same as pure, dry sand, as noted in the text (P8 L26 marked-up text).

The statement "Since much of the sediment heat content arises from SW insolation …" was not meant to imply that SW forcing is the dominant mechanism here – only that it can be important. This statement has now been reworded to reflect this (P13 L7-8, marked-up text).

***Minor Remarks:***

Response to Anonymous Referee #1

The noted references discussing the relationship between fetch and epilimnion depth are not intended to be comprehensive – merely indicative of the fact that it has long been known that a relationship exists (hence the older references).

Higher quality graphics can be provided at the production stage.

The issue of the overestimation of surface temperatures in both simulations during mid-day is indeed interesting and evidently little affected by a 50% reduction in surface stress. There is clearly information here indicating another process may be poorly simulated, but we have so far been unable to track this down.

**Incorporating Wind Sheltering and Sediment Heat Flux into 1-D Models of Small Boreal Lakes: A Case Study with the Canadian Small Lake Model V2.0**

Murray D. MacKay[1]

[1]Science and Technology Branch, Environment and Climate Change Canada, Toronto, M3H5T4, Canada

5  *Correspondence to*: Murray D. MacKay (murray.mackay@canada.ca)

**Abstract**  Lake models are increasingly being incorporated into global and regional climate and numerical weather prediction systems. Lakes interact with their surroundings through flux exchange at their bottom sediments and with the atmosphere at the surface, and these linkages must be well represented in fully coupled prognostic systems in order to completely elucidate the role of lakes in the climate system. In this study schemes for the inclusion of wind

10   sheltering and sediment heat flux simple enough to be included in any one dimensional lake model are presented. Example simulations with the Canadian Small Lake Model show improvements in surface wind driven mixing and temperature in summer, and a reduction of the bias in the change in heat content under ice compared with a published simulation based on an earlier version of the model.

15   *Copyright statement*.   The works published in this journal are distributed under the Creative Commons Attribution 4.0 License. This licence does not affect the Crown copyright work, which is re-usable under the Open Government License (OGL). The Creative Commons Attribution 4.0 License and the OGL are interoperable and do not conflict with, reduce or limit each other. © Crown copyright 2018

[revised manuscript text omitted]